# The causal role of auditory cortex in auditory working memory

**Liping Yu[†], Jiawei Hu[†], Chenlin Shi[†], Li Zhou, Maozhi Tian, Jiping Zhang, Jinghong Xu\***

Key Laboratory of Brain Functional Genomics (Ministry of Education and Shanghai), Key Laboratory of Adolescent Health Assessment and Exercise Intervention of Ministry of Education, and School of Life Sciences, East China Normal University, Shanghai, China

**Abstract** Working memory (WM), the ability to actively hold information in memory over a delay period of seconds, is a fundamental constituent of cognition. Delay-period activity in sensory cortices has been observed in WM tasks, but whether and when the activity plays a functional role for memory maintenance remains unclear. Here, we investigated the causal role of auditory cortex (AC) for memory maintenance in mice performing an auditory WM task. Electrophysiological recordings revealed that AC neurons were active not only during the presentation of the auditory stimulus but also early in the delay period. Furthermore, optogenetic suppression of neural activity in AC during the stimulus epoch and early delay period impaired WM performance, whereas suppression later in the delay period did not. Thus, AC is essential for information encoding and maintenance in auditory WM task, especially during the early delay period.

## Introduction

Working memory (WM) refers to the ability to actively hold information in memory over a time scale of seconds. It is a fundamental component of various cognitive functions (*Baddeley, 1992*). Previous studies have found that the prefrontal cortex (PFC) is crucial for WM because sustained neural activity was observed in PFC during the delay period of WM task (*Fuster and Alexander, 1971*; *Meyers et al., 2012*; *Miller et al., 1996*; *Romo et al., 1999*) and lesions to PFC produced profound WM deficits (*Petrides, 2000*). Besides, there is increasing evidence suggesting that sensory cortices are important components of the circuitry that underlies WM when the task requires short-term retention of sensory information (*Gayet et al., 2018*; *Pasternak and Greenlee, 2005*; *Scimeca et al., 2018*). For example, human fMRI studies had been successful in decoding WM information from the activity of sensory cortices, both visual (*Harrison and Tong, 2009*; *Serences et al., 2009*) and auditory (*Linke and Cusack, 2015*; *Linke et al., 2011*). Electrophysiological recordings in monkeys had observed memory-related activity during the delay period in sensory cortices in visual (*Supèr et al., 2001*), auditory (*Gottlieb et al., 1989*), and somatosensory (*Zhou and Fuster, 1996*; *Zhou and Fuster, 2000*) modalities. However, whether and when sensory cortices play a functional role in sensory memory maintenance in WM task remains unclear.

The auditory cortex (AC) is known to be a major site for auditory information processing. Some recent studies have reported that AC neurons also play a role in a number of auditory behavior tasks (*Fritz et al., 2003*; *Lee and Middlebrooks, 2011*; *Niwa et al., 2012*; *Otazu et al., 2009*). Furthermore, correlates of auditory WM have been reported in AC (*Bigelow et al., 2014*; *Scott et al., 2014*). Here, we delineate the functional role of AC in auditory WM.

Traditional methods for perturbing neural activity such as surgical lesion, pharmacological inactivation, and tissue cooling techniques do not provide the temporal resolution required for delineating the functional role of AC in memory maintenance in WM task because they disrupt both the

**\*For correspondence:**
jhxu@bio.ecnu.edu.cn

[†]These authors contributed equally to this work

**Competing interests:** The authors declare that no competing interests exist.

**eLife digest** Working memory is the ability to hold information in your head for a few seconds while making decisions, planning or applying logical reasoning to problem solving. It is a fundamental component of cognition, and yet it remains unclear where working memory is stored in the brain.

The prefrontal cortex – the front lobe of the brain – is likely the main hub of working memory, since it is responsible for executive functions, such as decision making and planning. This idea is supported by experiments showing sustained brain activity in the prefrontal cortex during working memory tasks. Lesions in that part of the brain also lead to profound deficits in working memory. However, there is increasing evidence that other parts of the brain which process sensory information also participate in retaining working memory. The auditory cortex, which processes sound, is one such candidate.

To find out whether the auditory cortex has a role to play in working memory, Yu, Hu, Shi et al. trained mice to lick a water spout after hearing the same sound twice in a row, 1.5 seconds apart, and then measured the activities of the mice's neurons. This showed that neurons in the auditory cortex were active not only when the mice were presented with sound cues, but also for a short time during the delay period between sounds. Yu, Hu, Shi et al. then manipulated this neurons to inactivate them for a fraction of a second after the first sound, which resulted in the animals' working memory was impaired. However, suppressing the activity of the auditory cortex cells in the later stages of the sound delay period had no effect on working memory.

These results indicate that although the auditory cortex may not be involved in storing information for the entire working memory process, it is crucial for encoding of auditory information. In summary, this work uncovers how neurons in the auditory cortex underlie working memory. Further research focusing on these neurons could explain how working memory deteriorates with age, or why it is impaired in people with learning difficulties.

encoding of the sensory stimuli and their retention in WM. Furthermore, the activity could not be silenced rapidly enough to test when AC contributes to memory maintenance during the delay period.

In the present study, we used a combination of optogenetics and electrophysiology methods to examine the functional role of AC in auditory WM. Because we were particularly interested to know when AC contributes to auditory memory maintenance, we applied optogenetic silencing at different time points across the delay period. We found that AC neurons exhibited elevated activity not only during the presentation of the auditory stimulus but also during the early delay period. Furthermore, optogenetic suppression of AC activity during the stimulus epoch and early delay period caused a reduction in WM performance, whereas suppression later in the delay period did not. These findings reveal a causal role of AC in encoding and maintaining the auditory information in the auditory WM task, especially early in the delay period.

## Results

### Behavioral performance in auditory WM task

We trained head-fixed mice to perform an auditory delayed match to sample (DMS) task (*Figure 1a*). In this task, water-restricted mice were presented a 200 ms auditory stimulus (3 kHz or 12 kHz tone) as the sample, followed by a delay period (1.5 s) and then a testing auditory stimulus, either matched or nonmatched to the sample. Licking within a response time window (1 s) in the match trial was rewarded with water (*Figure 1b*). Thus, mice had to remember the briefly presented auditory sample stimulus during the delay period. The behavioral performance declined with increasing duration of the delay period, indicating that the task required short-term memory processes (*Figure 1d*).

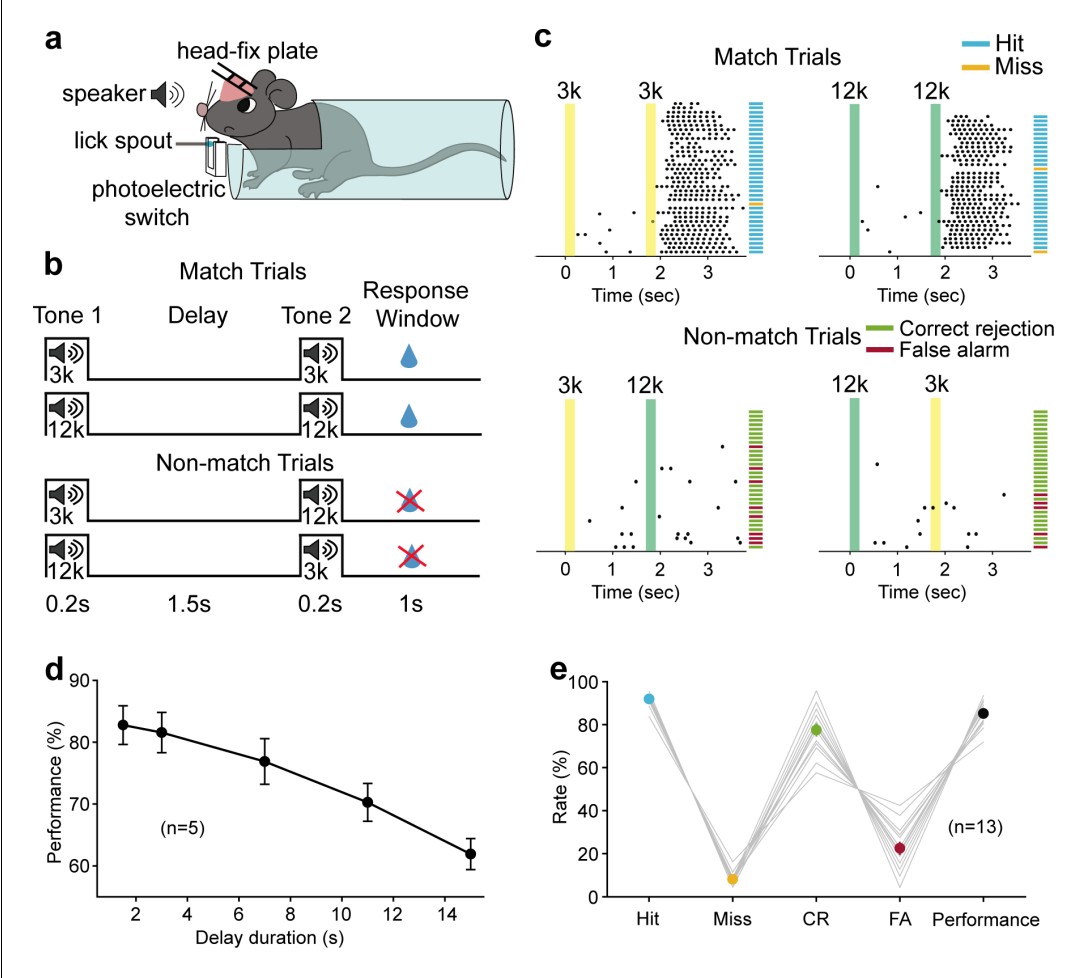

**Figure 1.** Auditory working memory task in head-fixed mice and behavioral performance. (**a**) Diagram of the experimental setup. (**b**) Schematic for task design. For each trial, an auditory stimulus (3 kHz, or 12 kHz, 0.2 s) was presented as the sample, followed by a delay period of 1.5 s and a testing auditory stimulus (0.2 s), either matched or nonmatched to the sample. Mice were rewarded with water if they licked within a response window in the match trials. (**c**) Licking behavior in an example session and definition of the trial type. Colored areas correspond to the two auditory stimulus delivery periods, as indicated above. Each tick indicates one lick. Short horizontal lines indicate the trial types (blue: hit; orange: miss; green: correct rejection [CR]; magenta: false alarm [FA]). (**d**) The performance with varying delay duration (n = 5 mice). Mean ± s.e.m. (**e**) Mean hit, miss, CR, FA rates, and the performance of all mice (n = 13 mice) during neural recording sessions. Gray lines: individual mice; black: mean ± s.e.m. Hit + miss = 100%; CR + FA = 100%. See *Figure 1—source data 1* for more details.

The online version of this article includes the following source data and figure supplement(s) for figure 1:

**Source data 1.** Performance in the auditory working memory task.

**Figure supplement 1.** The learning process of the auditory working memory (WM) task and the licking behavior of well-trained mice.

## Neural activity during auditory WM

To examine the neural correlate of auditory WM in AC, we recorded the single-unit activity of AC by using tetrodes while mice were performing the auditory DMS task (n = 13 mice, *Figure 1e*). A total of 915 neurons were recorded, of which 287 (31.4%) exhibited activity related to the task (compared with baseline activity, evaluated with a paired *t* test, at the p<0.05 level) and were selected for further analysis. Responses from one typical AC neuron are shown in *Figure 2a* left. The neuron exhibited a phasic response during the auditory sample stimulus presentation. After the offset of the sample stimulus, the neuron continued to exhibit activity for 800 ms into the delay period. This response pattern was representative of our population (*Figure 2b*). *Figure 2c* shows that more than 60% of AC neurons exhibited increased firing rate during the first 500 ms of the delay. The incidence

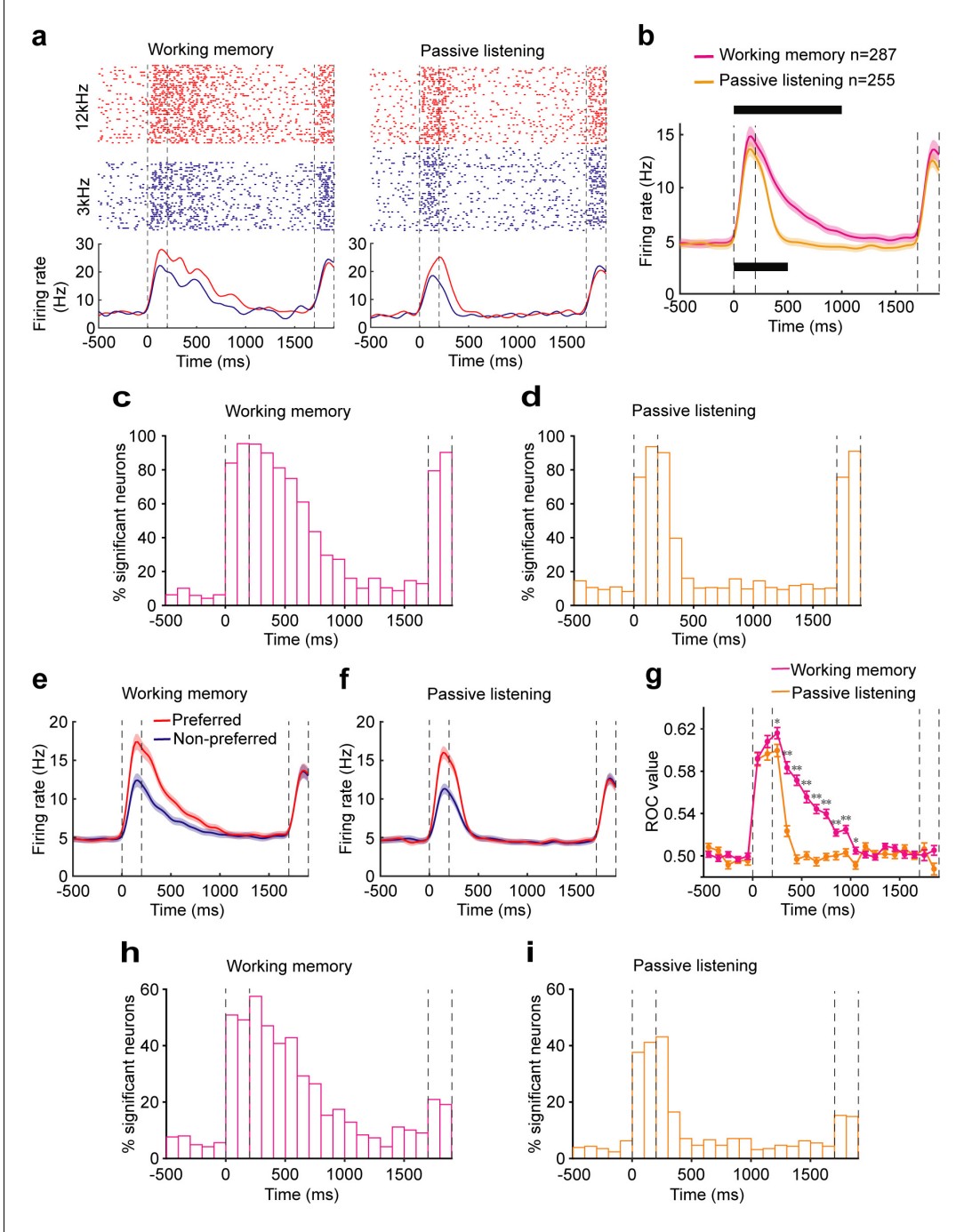

**Figure 2.** Neural correlates of the auditory cortex activity in the auditory working memory (WM) task. (a) Raster (top) and peri-stimulus time histograms (bottom) of an example neuron recorded during the WM behavior (left) and passive listening (right). Trials were sorted by auditory samples. The sample stimulus and test stimulus times are bounded by the vertical dotted lines. (b) Averaged population firing rates for neurons recorded during WM behavior (n = 287) and passive listening (n = 255). Shadows: s.e.m.; the black block on the top indicates the successive 100 ms bins with firing rate significantly different from baseline (500 ms before the beginning of sample) for neurons recorded during WM behavior. p<0.05, Wilcoxon rank-sum test. The black block below indicates significant bins for neurons recorded during passive listening. (c, d) Percentage of neurons with a significant difference in firing rate compared with baseline at different time points during WM behavior (c) and passive listening (d). (e, f) Averaged population firing rates for neurons recorded during WM behavior (e) and passive listening (f). Trials in which the sample stimulus was the preferred or nonpreferred, which varied for each neuron, are shown separately. (g) The average of receiver operating characteristic (ROC) values across populations, calculated in each 100 ms window, is plotted as a function of time for neurons recorded during WM behavior (magenta) and passive listening (orange).

*Figure 2 continued on next page*

*Figure 2 continued*

*p<0.05, **p<0.001, Wilcoxon rank-sum test. (**h, i**) Incidence of neurons with significant ROC values for each 100 ms epoch in WM behavior (**h**) and passive listening (**i**). p<0.05, permutation test.
The online version of this article includes the following figure supplement(s) for figure 2:

**Figure supplement 1.** Neural correlates of the auditory cortex activity in the auditory working memory (WM) task with varied stimulus duration.

decreased to ~30% during 500–800 ms of the delay. For the majority of these units (~90%), the elevated firing rate appeared in the first 800 ms of the delay period and did not persist any further.

As shown in the example (*Figure 2a* left) and population (*Figure 2e*), the firing rate of AC neurons showed selectivity of the preceding sample stimulus during both the auditory stimuli and early delay. To quantify the ability to discriminate between the two stimuli, we performed a receiver operating characteristic (ROC) analysis. The mean ROC values for the population of AC neurons increased rapidly after the onset of the auditory stimulus and retained above the value of 0.5 during the early delay (<800 ms) (*Figure 2g*). The permutation test also showed that about 50% of neurons showed significant sample selectivity during the stimulus and early delay (500 ms after stimulus offset) (p<0.05). The proportion decreased to 20–30% during the subsequent period of 500–800 ms and dropped to about 10% after 800 ms (*Figure 2h*). These results indicated that AC neurons carried the auditory information during the early delay period in WM task.

To determine whether this delay-period activity was related to the task or only reflected the continuation of the auditory response, we recorded 836 neurons during the passive presentation of the stimulus. 255 neurons responded to at least one auditory sample stimulus (compared with baseline activity, paired *t*-test, p<0.05). In the passive presentation condition, AC neurons exhibited robust responses and showed stimulus selectivity during the sample. They showed selectivity for a much shorter interval in the delay period, in contrast to results obtained during WM task (*Figure 2*). These results indicated that the continuation of neural activity into the delay period in WM task was not a passive process but was related to the animals' behavior.

We observed that after the offset of the 200 ms sample stimulus, the neuron continued to exhibit activity for 800 ms into the delay period, 1000 ms from the stimulus onset in total. To test whether the temporal dynamics of delay-period activity is relative to the stimulus onset or offset, we increased the stimulus duration to 300 ms and 400 ms and test whether the sustained activity would shift with later stimulus offset. The AC activity was recorded from a subgroup of mice (n = 4) while performing the auditory WM task with the stimulus duration of 300 ms or 400 ms. The averaged population firing rates showed that the neurons exhibited phasic responses during the auditory sample stimulus presentation. After the offset of the sample stimulus, the neurons continued to exhibit elevated activity for 700 ms into the delay period in the 300 ms stimulus duration task and exhibit elevated activity for 600 ms into the delay period in the 400 ms stimulus duration task (*Figure 2— figure supplement 1*). Together with the result from the 200 ms sample stimulus duration task, which showed that the neurons continued to exhibit elevated activity for 800 ms into the delay period, all of these results showed that AC neurons exhibited elevated activity for 1000 ms from the stimulus onset, regardless of the duration of the sample stimulus. These results suggested that the temporal dynamics of the delay-period activity in AC might be relative to the stimulus onset rather than the offset.

## Early delay activity of AC is required for memory maintenance in auditory WM task

To directly test whether and when AC contributes to auditory WM, we suppressed the activity of AC pyramidal neurons transiently during the delay period with optogenetic methods. It was achieved through expressing the inhibitory halorhodopsin (eNpHR3.0) in pyramidal neurons by injecting AAV-CaMKIIα-eNpHR3.0-eYFP into AC. The expression and functionality of NpHR were verified by immunostaining and optetrode recording (*Figure 3a, b*). We applied optogenetic suppression during the delay period by randomly interleaving 'laser ON' and 'laser OFF' trials in the same behavioral session. Suppressing the activity of AC pyramidal neurons during the delay period caused a strong impairment in task performance (*t*-test, p=1.1 × $10^{-5}$, laser ON: 64.84 ± 1.59%; laser OFF: 77.98 ± 1.17%, n = 8), with a substantial increase in false alarm rate (*t*-test, p=2.63 × $10^{-5}$, laser ON: 50.96 ±

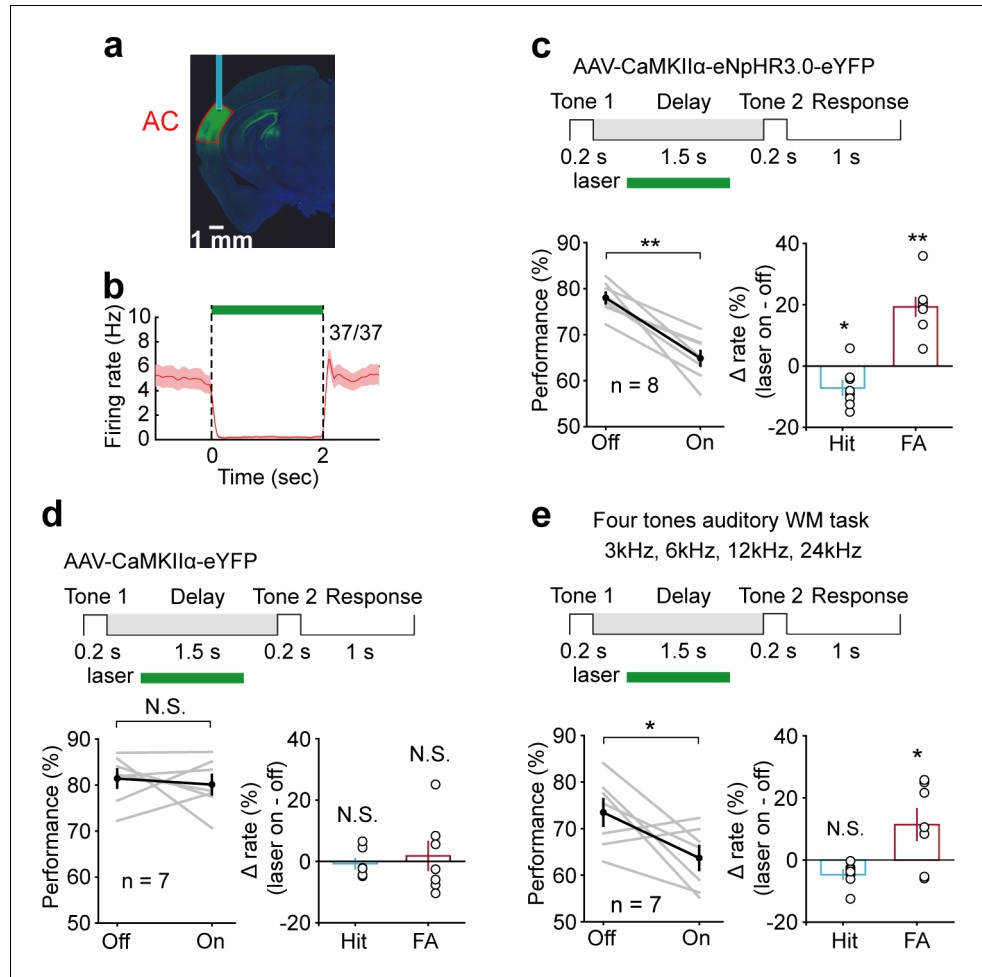

**Figure 3.** Suppression of delay-period activity in auditory cortex (AC) by optogenetic inhibition of pyramidal neurons impaired auditory working memory (WM) performance. (a) Histology image showing the expression of AAV-CaMKIIα-eNpHR3.0-eYFP in AC. (b) Activity suppression efficiency revealed by optetrode recording in vivo. (c) Suppressing AC activity during the delay period of WM task decreased performance, with a substantial increase in false alarm (FA) rate and a small decrease in hit rate. Top: schematic of optogenetic stimulation during the delay period of WM task. The green rectangle indicates the period of inactivation. For the bottom panel, gray lines indicate individual mice; black indicates mean ± s.e.m. Circles indicate individual mice. *p<0.05, **p<0.001, *t*-test. (d) As in (c) with control virus injection. The photostimulation of AC with control virus injection during the delay period did not affect the behavior. N.S.: not significant. (e) Suppressing AC delay-period activity decreased the performance in the four tones auditory WM task with a decrease in FA rate and no change in hit rate. See *Figure 3—source data 1* for more details.

The online version of this article includes the following source data and figure supplement(s) for figure 3:

**Source data 1.** Effect of auditory cortex suppression on working memory behavior.

**Figure supplement 1.** Optogenetic suppression of auditory cortex during the stimulus epoch dramatically reduced the animals' ability to perform the working memory task.

2.65%; laser OFF: 31.66 ± 1.71%, n = 8) and a small decrease in hit rate (*t*-test, p=0.018, laser ON: 80.52 ± 1.74%; laser OFF: 87.69 ± 2.04%, n = 8) (*Figure 3c, d*). Thus, the delay-period activities of AC pyramidal neurons were important for maintaining the auditory information after the auditory stimulus ceased.

Next, we test the necessity of AC during the stimulus epoch. Optogenetic suppression of AC during the stimulus epoch dramatically reduced the animals' ability to perform the task (*t*-test, p=4.92 × 10$^{-7}$, laser ON: 61.71 ± 2.12%; laser OFF: 81.76 ± 1.3%, n = 9), (*Figure 3—figure*

*supplement 1*). This result indicated that AC was crucial for the initial encoding of auditory information in the WM task.

For a genuine WM task, subjects should be able to perform beyond two cues. To further test the role of AC in auditory WM with more cues, a subgroup of mice (n = 7) was tested in a four tones auditory WM task (3 kHz, 6 kHz, 12 kHz, 24 kHz). Again, optogenetic suppression of AC during the delay period resulted in substantial impairment in task performance (*t*-test, p=0.025, laser ON: 63.69 ± 2.58%; laser OFF: 73.48 ± 2.84%, n = 7) (*Figure 3e*).

Electrophysiology results showed that AC neurons continued to exhibit activity for 800 ms into the delay period, and these activities did not persist any further. We speculated that AC neurons might carry auditory information only during this early delay period. Therefore, optogenetic suppression of AC would be expected to affect WM only when applied during this early delay period. To test this idea, we divided the delay into two epochs, 300–800 ms and 800–1300 ms, and applied optogenetic suppression briefly within each epoch individually. Notably, optogenetic suppression during the epoch of 300–800 ms caused a strong reduction in behavioral performance (*t*-test, p=0.01, laser ON: 70.36 ± 2.5%; laser OFF: 80.51 ± 1.58%, n = 8) (*Figure 4a*). Optogenetic suppression during the epoch of 800–1300 ms produced a smaller effect that was not significantly different from the laser-off condition (*t*-test, p=0.34, laser ON: 81.68 ± 1.72%; laser OFF: 83.91 ± 1.47%, n = 8) (*Figure 4b*). To examine the temporal specificity of the effect of AC suppression more precisely, we used even shorter inactivation periods: 300–550 ms, 550–800 ms, 800–1050 ms, and 1050–1300 ms. Again, only inactivation during the 300–550 ms and 550–800 ms had an effect on behavioral performance (300–550 ms: *t*-test, p=0.016, laser ON: 73.11 ± 2.23%, laser OFF: 79.38 ± 1.4%, n = 8; 550–800 ms: *t*-test, p<0.001, laser ON: 73.72 ± 2.82%, laser OFF: 86.31 ± 1.13%, n = 8), while the effect of inactivation after 800 ms was small and not statistically significant (800–1050 ms: *t*-test, p=0.38, laser ON: 77.93 ± 2%, laser OFF: 80.68 ± 2.25%, n = 7; 1050–1300 ms: *t*-test, p=0.42, laser ON: 84.97 ± 1.57%, laser OFF: 82.74 ± 2.19%, n = 8) (*Figure 4c–e*).

To further examine whether the temporal specificity of the AC suppression effect changes with increasing delay length, we increased the delay length to 3 s and 7 s. Again, only perturbation during 300–800 ms had a significant disruption on behavior (3 s delay, *t*-test, p=1.16 × 10$^{-3}$, laser ON: 71.69 ± 1.79%, laser OFF: 81.81 ± 1.73%, n = 8; 7 s delay, *t*-test, p=0.033, laser ON: 72.8 ± 2.13%, laser OFF: 80.38 ± 2.32%, n = 7) (*Figure 5a, c*). No obvious effects were seen when optogenetic suppression occurred after 800 ms (3 s delay, *t*-test, p=0.39, laser ON: 82.53 ± 1.92%, laser OFF: 85.33 ± 2.51%, n = 8; 7 s delay, *t*-test, p=0.17, laser ON: 81.25 ± 1.83%, laser OFF: 84.69 ± 1.49%, n = 7) (*Figure 5b, d*). These results indicated that AC is critical for WM only during the early delay period.

Active WM maintenance requires resistance to distractors presented during the delay period. To test this ability in mice, we added a noise distractor (20–20,000 Hz, 200 ms, 60 dB) during the early (300–500 ms) delay period of the WM task. Mice were first trained to perform the WM task, and then the noise distractor was added. Mice could quickly adapt to the WM task with noise distractor, despite the initial drop in performance. After 2 days of training, the performance on the third day of the WM task with noise distractor was no worse than that in the simple WM task (*t*-test, p=0.247, WM task with noise distractor: 79.55 ± 1.77%; simple WM task: 82.4 ± 1.58%, n = 9) (*Figure 6a*). We then optogenetically suppressed the activity of AC pyramidal neurons after the noise distractor in the WM task. Optogenetic suppression of AC during the early delay period after the distractor (500–800 ms) resulted in impairment in task performance (*t*-test, p=0.017, laser ON: 69.83 ± 2. 8%; laser OFF: 77.32 ± 1.63%, n = 9), with a substantial increase in false alarm rate (*t*-test, p=0.016, laser ON: 45.6 ± 4.48%; laser OFF: 33.49 ± 2.5%, n = 9) and no significant change in hit rate (*t*-test, p=0.156, laser ON: 85.29 ± 1.68%; laser OFF: 88.41 ± 1.26%, n = 9) (*Figure 6b*). Thus, AC activity is important for active early maintenance of the auditory information in the face of noise distractor in the WM task.

To study the functional specificity of optogenetic suppression, we designed an additional series of experiments. First, we trained another group of mice to perform a delayed go/no-go auditory discrimination task. The stimuli were the same as the DMS task, except that mice made a decision depending on the first stimulus (*Figure 7a*). This task required auditory perception, the memory of the task-relevant information, but not auditory frequency information retention during the delay period (*Goard et al., 2016*; Kamigaki T and *Kamigaki and Dan, 2017*). Laser illumination was applied after the sample stimulus and before the go signal, simulating the delay period in the DMS task. We found that laser illumination had no effect on the behavior (*t*-test, p=0.335, laser ON: 76.81

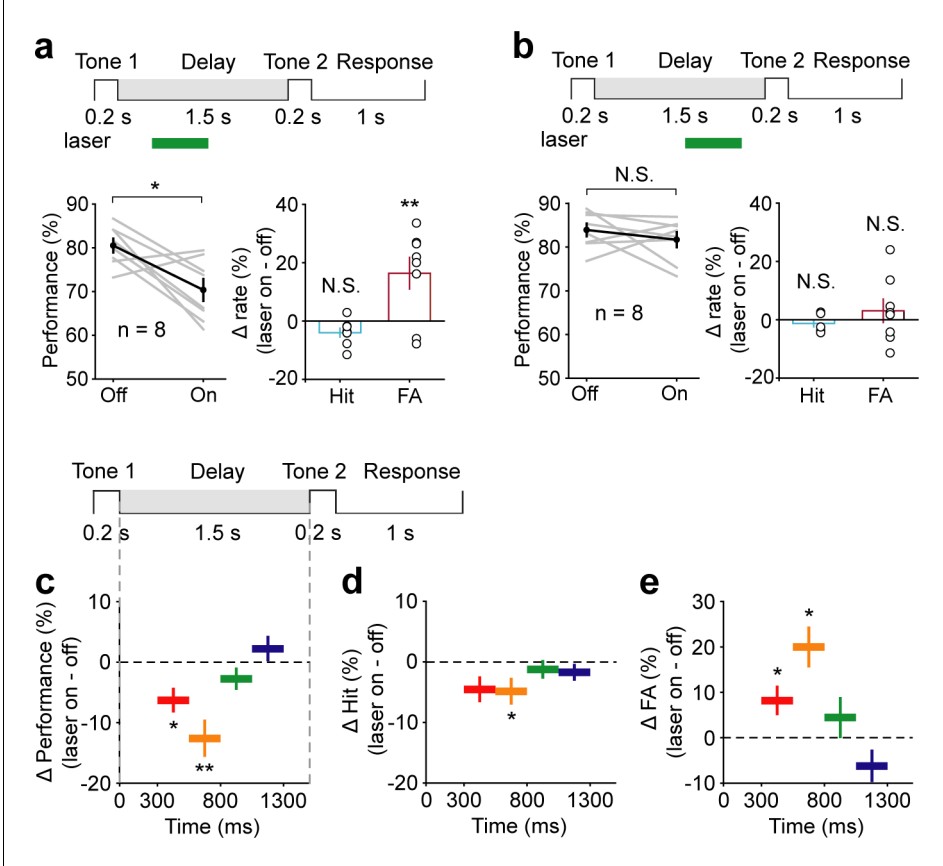

**Figure 4.** Temporal specificity of the effect of auditory cortex (AC) suppression. (a) AC suppression during the delay period of 300–800 ms decreased the working memory (WM) behavioral performance, with an increase in false alarm (FA) rate and no change in hit rate. (b) AC suppression during the delay period of 800–1300 ms did not affect behavioral performance. (c–e) The WM behavioral change caused by AC suppression during the delay period of 300–550 ms (red; n = 8 mice), 550–800 ms (yellow; n = 8 mice), 800–1050 ms (green; n = 7 mice), and 1050–1300 ms (blue; n = 8 mice). The task structure is shown at the top. For the bottom panels, the horizontal extent of the colored bars indicates the period of inactivation. The vertical position indicates the average change in performance (c), hit rate (d), and FA rate (e) across mice induced by the corresponding period of AC suppression. Error bars show s.e.m. across mice. See *Figure 4—source data 1* for more details.
The online version of this article includes the following source data for figure 4:

**Source data 1.** Effect of auditory cortex suppression during delay period of 300–800 ms and 800–1500 ms.

± 2.01%, laser OFF: 79.9 ± 2.34%, n = 8) (*Figure 7a*). Therefore, AC delay-period activity appeared to be more important in memory of the auditory cue but not the action plan. Second, mice were trained to perform a go/no-go auditory discrimination task. The stimuli were the same as the DMS task, except that mice made a decision based on the second stimulus (*Figure 7b*). This task retained the general processes such as attention or expectation of the forthcoming second stimulus but did not require auditory WM during the delay period. Laser illumination was applied after the starting cue and before the sample stimulus, simulating the delay period in the DMS task. The result showed that laser illumination produced a smaller effect that was not significantly different from the laser-off condition (Wilcoxon rank-sum test, p=0.536, laser ON: 81.55 ± 3.1%, laser OFF: 84.71 ± 2.2%, n = 9) (*Figure 7b*). The effect of AC suppression in go/no-go task is much smaller than that in the DMS task (t-test, p=7.68×10$^{-3}$, Δperformance [laser ON-laser OFF], go/no-go = −3.15 ± 2.5%, n = 9; DMS = -13.14 ± 1.98%, n = 8). Therefore, AC delay-period activity appeared to be more critical in memory retention than other general processes such as attention or expectation of the forthcoming second stimulus.

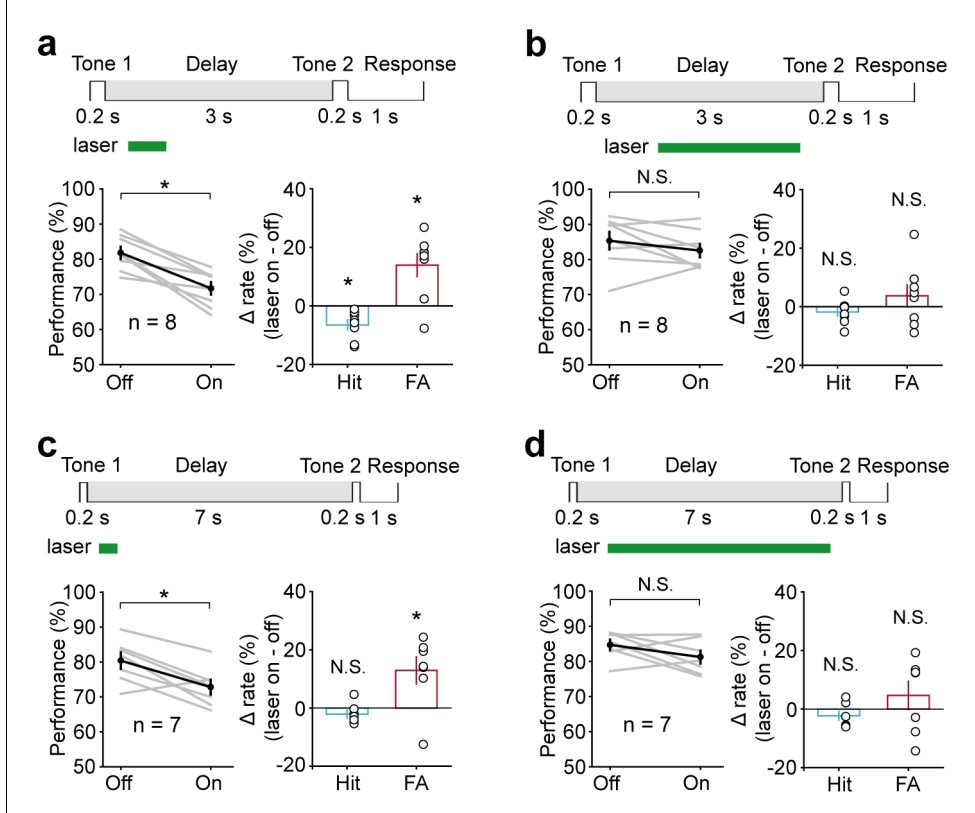

**Figure 5.** Performance was impaired following optogenetic suppression of auditory cortex (AC) activity during the early delay period, with the delay duration of 3 s and 7 s. (**a, b**) In working memory (WM) task with the delay duration of 3 s, AC suppression during the early delay period (300–800 ms) (**a**) but not later (800–2700 ms) (**b**) decreased the behavioral performance. (**c, d**) In WM task with the delay duration of 7 s, AC suppression during the delay period of 300–800 ms (**c**) but not 800–6700 ms (**d**) decreased the behavioral performance. See *Figure 5— source data 1* for more details.

The online version of this article includes the following source data for figure 5:

**Source data 1.** Effect of auditory cortex suppression in working memory task with the delay duration of 3 s and 7 s.

## Discussion

By measuring and manipulating the activity of neurons in AC during the auditory WM behavior, we explored the causal role of AC in WM. Experiments showed that AC neurons were active not only during the presentation of the auditory stimulus but also during the early delay period when no stimulus was presented. Further experiments showed that optogenetic suppression of neuronal activity in AC during the stimulus epoch and early delay period caused a substantial reduction in WM performance, whereas suppressing later in the delay period did not. These results indicated that although AC may not be involved in WM storage during the whole delay period, it is crucial for WM tasks in the initial encoding of auditory information, maintaining the memory trace for a limited time, and then transferring this information for further WM storage elsewhere.

The role of sensory cortices in WM is debate. There is evidence supporting the idea that the same system involved in sensory processing also participates in retaining sensory information in WM task (*Gayet et al., 2018*; *Pasternak and Greenlee, 2005*; *Scimeca et al., 2018*). It is based on fMRI decoding and monkey neurophysiology studies showing that during the memory period neuronal activity in sensory cortices can be maintained and reflected the identity of the remembered stimulus (*Fuster and Jervey, 1981*; *Harrison and Tong, 2009*; *Mendoza-Halliday et al., 2014*). Furthermore, perturbation of neural activity in sensory cortices can impair WM performance (*Colombo et al., 1990*; *Harris et al., 2002*; *Zhang et al., 2019*). However, there are also studies showing that when the distractor is applied during the delay period, the sustained neural activity and WM decoding

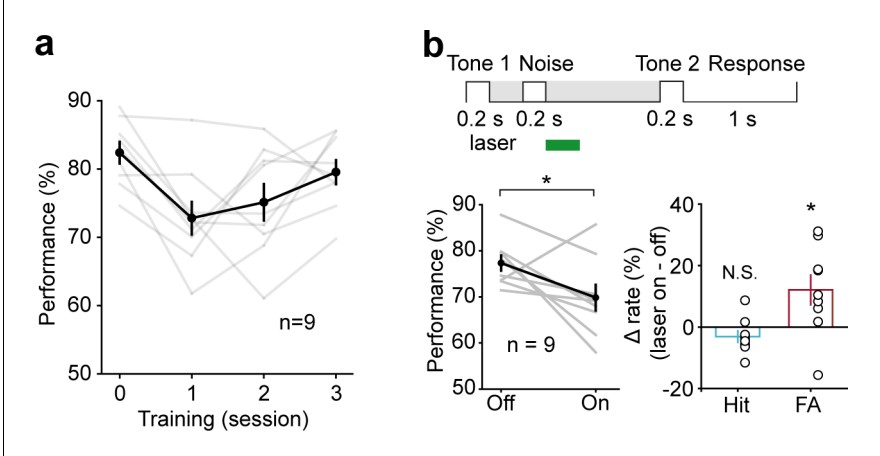

**Figure 6.** Active memory maintenance in auditory working memory (WM) task by the auditory cortex (AC) delay-period activity. (a) Learning curve for the performance in the WM task with noise distractor (presented during 300–500 ms of the delay period). Note the drop of performance after inserting the noise distractor in the delay period on the first day. After 2 days of training, the performance data from the third day of the WM task with noise distractor was no worse than that in the simple WM task (the zeroth day). (b) Optogenetic suppression of AC during the early delay period after the distractor (500–800 ms) resulted in impairment in task performance. The online version of this article includes the following source data for figure 6:

**Source data 1.** Effect of auditory cortex suppression in working memory task with noise distractor.

were no longer present in sensory cortices. Meanwhile, the presence of distractor had no impact on behavioral performance (*Bettencourt and Xu, 2016*; *Miller et al., 1996*). This suggests that sensory cortices may not be essential for memory maintenance in WM task and WM-related activities observed in sensory cortices may largely reflect feedback signals indicative of the storage of WM

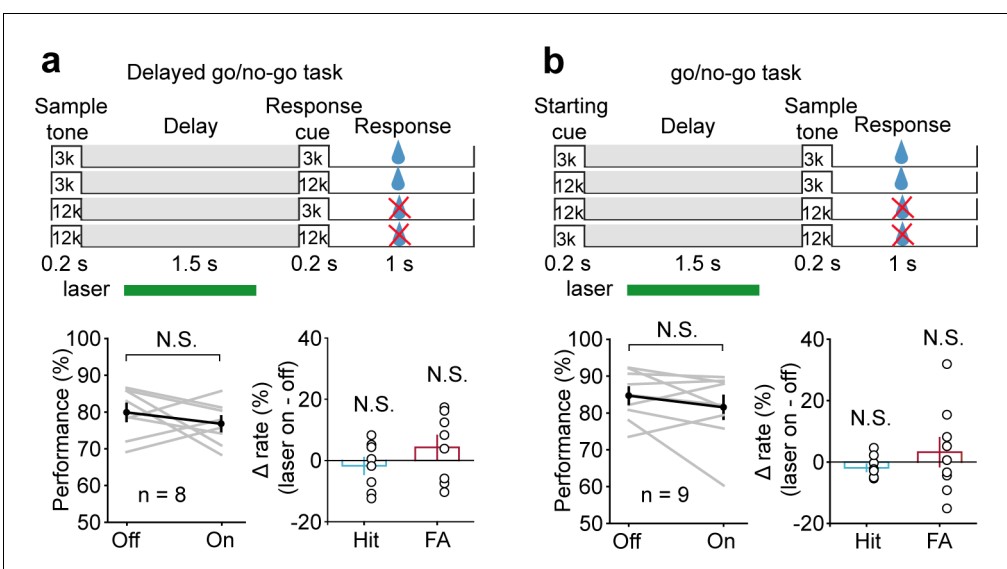

**Figure 7.** Suppressing auditory cortex (AC) activity did not affect behavioral performance in the delayed go/no-go auditory discrimination task and go/no-go auditory discrimination task. (a) Paradigm and behavioral performance for the delayed go/no-go auditory discrimination task experiments with suppressed AC activity. (b) As in (a) for the go/no-go auditory discrimination task. See *Figure 6—source data 1* for more details.
The online version of this article includes the following source data for figure 7:

**Source data 1.** Effect of auditory cortex suppression in delayed go/no-go auditory discrimination task and go/no-go auditory discrimination task.

content elsewhere in the brain, probably in the posterior parietal cortex and PFC (*Leavitt et al., 2017*; *Xu, 2017*; *Xu, 2018*).

Active WM maintenance requires resistance to distractors presented during the delay period (*Baddeley, 2012*; *Bettencourt and Xu, 2016*; *Miller et al., 1996*). If the delay-period activity of a sensory cortex is essential for WM, the neural activity should be able to maintain information even following the distractor. With regard to the present study, if suppressing the delay-period activity of AC following the distractor impairs the performance, it will implicate AC in active information maintenance in WM task. As we expected, our result showed that suppressing the early delay-period activity of AC after the distractor resulted in performance impairment. Therefore, AC is essential for active WM maintenance even in the face of the distractor. The early delay-period activity observed in AC reflected the active memory maintenance, not the residual auditory trace. Thus, our present results support the idea that sensory cortices participate in sensory information maintenance in WM task.

In the present study, we observed a temporal-specific effect of AC suppression during the delay period of WM task. Only optogenetic suppression of AC activity during the stimulus epoch and early delay period (300–800 ms) caused a substantial reduction in WM performance. These results were consistent with the electrophysiology results showing that AC neurons exhibited elevated activity during stimulus presentation and the first 800 ms of the delay period. These findings agree with transcranial magnetic stimulation (TMS) studies in humans showing that disruption of primary somatosensory cortex functioning early in the delay period (at 300 ms or 600 ms) interfered with tactile WM performance. In contrast, TMS later in the delay did not (*Harris et al., 2002*).

Furthermore, the WM performance impairment induced by delay-period optogenetic suppression of AC cannot be attributed to direct disruption of auditory perception. Because AC suppression during the delay period of a delayed go/no-go auditory discrimination task that also required auditory sample stimulus perception had no effect on the behavior (*Figure 7a*). It is also unlikely that the optogenetic suppression during the delay period impaired perception of the test stimulus. Because laser illumination during the delay period of 800–1300 ms, before the test stimulus, did not affect WM behavior (*Figure 4b*).

In contrast to the result of the DMS task, we found that suppression of AC activity during the delay period of a delayed go/no-go task did not affect behavior. There is an essential difference in the memory requirement for the two tasks. While the DMS task requires short-term memory of the sensory cue, the delayed go/no-go task is likely to require maintenance of the action plan. This result suggests that the sensory cortex is essential in memory of the sensory cue but not the action plan. This result was also in accord with a recent study finding that the activity of the visual cortex was necessary for the encoding of the stimulus but not for maintenance of the action plan in the memory-guided visual discrimination task (*Goard et al., 2016*).

Much of the information about the neural substrates for sensory WM comes from studies of the visual system in nonhuman primates. Relatively fewer studies deal with the storage of information in the auditory modality, perhaps in part due to the difficulties associated with training nonhuman primates to perform auditory tasks (*Bigelow et al., 2014*; *Cohen et al., 2005*; *Fritz et al., 2005*; *Munoz-Lopez et al., 2010*; *Scott et al., 2012*). In the present study, we have taken advantage of the relative ease and speed with which rodents can be trained on auditory DMS task to invest the causal role of AC in auditory WM. Future studies are needed to clarify the similarities and differences in neural circuitry and functional principle of auditory and visual WM.

WM-related activities during the delay period of WM task have been observed in distributed cortical and subcortical structures, including the PFC (*Fuster and Alexander, 1971*; *Goldman-Rakic, 1996*; *Miller et al., 1996*; *Romo et al., 1999*; *Sreenivasan et al., 2014*; *Ungerleider et al., 1998*), parietal cortex (*Chafee and Goldman-Rakic, 1998*; *Harvey et al., 2012*; *Shadlen and Newsome, 2001*), and superior colliculus (*Kopec et al., 2015*). The relative importance of these different regions in auditory WM remains to be investigated. Furthermore, these regions are anatomically connected to the AC. Such inter-areal interactions are likely to be essential for memory maintenance in the WM task. An important future direction would be to examine the neural circuits underlying WM.

# Materials and methods

**Key resources table**

| Reagent type (species) or resource | Designation | Source or reference | Identifiers | Additional information |
|---|---|---|---|---|
| Strain, strain background (*Mus musculus*) | C57BL/6 | Slac Laboratory Animal | N/A | |
| Other | Formvar-Insulated Nichrome Wire | A-M Systems | 761000 | |
| Other | Head-stage amplifier | Intan Technology | RHD2132 | |
| Software, algorithm | MATLAB | MathWorks | SCR-001622 | |

## Animals

Adult male C57BL/6 mice, aged 8–12 weeks at the start of the experiment, were used for this study. All experiments were performed in strict accordance with the recommendations in the Guide for the Care and Use of Laboratory Animals of the US National Institutes of Health. The protocol was approved by the Animal Care and Use Committee of East China Normal University, Shanghai, China. After the start of behavioral training, mice were placed on a restricted water schedule. Water was available only during task performance and immediately after the task. Each mouse's body weight was measured to ensure that weight loss did not exceed 20% of pre-water restriction weight. Sample sizes were similar to others used in the field. No statistical method was used to predetermine sample size.

## Behavior

### Behavioral system

Behavioral training was conducted in a custom-designed double-walled box covered with polyurethane foam for sound attenuation. Mice were head-fixed and placed in a polypropylene tube to limit movement (*Figure 1a*). Behavioral training and testing were implemented with custom software written in MATLAB (MathWorks). The sound signal generated by the computer is fed forward to analog-digital multifunction card (DAQ NI 6363, National Instruments, Austin, TX, USA), amplified (SA1, Tucker-Davis Technologies, FL, USA), and sent to the speaker (MF1, Tucker-Davis Technologies) located 20 cm in front of the mice. Licking signals were detected using a photoelectric switch mounted on either side of the lick spout, digitized by an analog-digital multifunction card (DAQ NI 6363, National Instruments), and saved in the computer. A small overhead camera with a microphone allowed audiovisual observation by the experimenter.

### Auditory DMS task

We trained head-fixed mice to perform an auditory DMS task (*Figure 1b*). Each trial began with a sample tone stimulus (3 kHz or 12 kHz, 60 dB, 200 ms), followed by a 1.5 s delay period, after which a test tone stimulus, same to (match) or different from (nonmatch) the sample, was presented. Mice were trained to lick the spout in match trials and withhold in nonmatch trials in the response window. The response window started immediately after the offset of the test tone stimulus and lasted 1 s. An inter-trial interval of 10 s separated successive trials. Responses were marked as follows: (1) hit: licking events were detected in the response window in match trials, and water delivery was triggered instantaneously. (2) Correct rejection: no licking was detected in the response window in nonmatch trials. (3) False alarm: licking events were detected in the response window in nonmatch trials. (4) Miss: no lickings were detected in the response window in match trials. A reward of ~5 µl water was triggered immediately only after hit; mice were neither punished nor rewarded for correct rejection, false alarm, or miss trials. The performance is calculated as follows:

$$Performance = (H + CR)/(H + CR + FA + M)$$

where H, CR, FA, and M are the numbers of hit, correct rejection, false alarm, and miss trials, respectively.

Hit and false alarm rates are defined as follows:

$$Hit\,rate = H/(H+M)$$

$$False\,alarm\,rate = FA/(FA+CR)$$

Mice were trained in three successive stages to achieve the performance >80%: (1) Mice were trained to lick the spout for water (2–3 days). (2) Only match trials were applied, and water was delivered only during the response window in all the trials. In the initial 2–3 days of this stage, water was delivered manually during the response window. The goal of this stage was simply to teach the mice to link the water reward with the serial of stimulus and encourage the mice to lick the spout during the response window. In the late part of this stage (3–4 days), water was delivered only when the licking events were detected during the response window. The goal of this stage was to check whether mice could lick the spout in the response window spontaneously. (3) Once mice could lick water in the response window in 90% of the trials, the two match (3 kHz–3 kHz, 12 kHz–12 kHz) and two nonmatch trials (3 kHz–12 kHz, 12 kHz–3 kHz) were presented randomly. Mice were only rewarded for licks during the response window in the match trials. The level of behavioral performance reached >80% correct after 15–24 days (on average 20 days) of training in this stage.

Totally, it took an average of 29 (25-35) days of training to reach the level of behavioral performance >80% correct. The performances of all mice during learning are shown in *Figure 1—figure supplement 1a*. In total, we trained 58 mice to perform the auditory WM task and removed 17 mice from the study before data collection because they failed to show apparent progress like others in stage 3 of training during which the two match and two nonmatch trials were presented randomly. Finally, the majority (71%, 41/58) of mice could learn the task.

All data described in this study were collected from well-trained mice. After behavioral performance reached >80% for more than three consecutive days, we started the main experiments, including electrophysiology and optogenetic experiments.

In the increasing delay duration experiments, mice were first trained with the delay period of 1.5 s. After the mice were well trained, they were tested with the delay period of different durations in 1 day. An inter-trial interval twice as long as the delay duration was used to separate successive trials.

In the 3 s and 7 s delay duration optogenetic inactivation experiments, mice were first trained to perform the DMS task with 1.5 s delay to the well-trained criterion. Then the delay duration was increased to 3 s or 7 s. After mice were well trained with the 3 s or 7 s delay task, we started the optogenetic inactivation experiments.

In the four tones optogenetic inactivation experiments, 3 kHz, 6 kHz, 12 kHz, and 24 kHz pure tones were used for match or nonmatch tone pairs. Mice were firstly well trained with the two tones DMS task (3 kHz and 12 kHz) and then trained with the four tones DMS task. After the mice were well trained with the four tones task, we started the optogenetic inactivation experiments.

For the optogenetic experiments with the blind design (*Figure 3c, d*), JH Xu labeled containers for the virus with 'A' and 'B'. She did not participate in behavioral and optogenetic experiments or data analysis.

## Delayed go/no-go auditory discrimination task

A separate group of mice were trained to perform a delayed go/no-go auditory discrimination task. Each trial consisted of a sample tone, delay, and response cue. During the sample tone period (200 ms), a target (3 kHz) or nontarget (12 kHz) tone stimulus was presented (the corresponding trial was referred to as a 'go' trial or 'no-go' trial, respectively), followed by a 1.5 s delay period. After the response cue was presented, licking in 'go' trials within the 1 s response window was rewarded (hit). No licking in response to a target or nontarget tone in the response window was regarded as miss or correct rejection (CR), respectively. Licking to a nontarget tone in the response window was regarded as false alarm (FA). The intertrial interval was 10 s. The performance, hit, and false alarm rates were calculated as described in the DMS task.

## Go/no-go auditory discrimination task

In the go/no-go auditory discrimination experiments, each trial consisted of a starting cue, delay, and sample tone. Trials began with the presentation of the starting cue, followed by a 1.5 s delay period. A sample tone stimulus was subsequently presented. The sample tone was either a target tone (3 kHz, 'go') or a nontarget tone (12 kHz, 'no-go'). Licking in response to a target tone in the response window (1 s) was rewarded (hit). No licking in response to a target or nontarget tone in the response window was regarded as miss or correct rejection (CR), respectively. Licking to a nontarget tone in the response window was regarded as false alarm (FA). The intertrial interval was 10 s. The performance, hit, and false alarm rates were calculated as described in the DMS task.

The mean lick peri-stimulus time histograms (PSTHs) of well-trained mice while performing the WM task delayed go/no-go auditory discrimination task, and go/no-go auditory discrimination task are shown in *Figure 1—figure supplement 1*. Once the mice were well trained, they readily performed the task with little licking during the delay period, and these lickings were not punished.

## Electrophysiology

### Assembly of tetrodes and optetrodes

Tetrodes were fabricated by twisting four Formvar-Insulated Nichrome Wire (bare diameter: 17.78 μm, A-M Systems, WA, USA) together. To construct a tetrode, a 20-cm-long wire was folded in half for twice over a horizontal bar. The end was clamped together with a clip and manually twisted clockwise. Finally, insulation coats of wires were gently heated to fuse with a heat gun, and the tetrode tips were cut. To reinforce each tetrode longitudinally, each tetrode was then inserted into a polymide tubing (inner diameter: 114.3 μm; wall: 12.7 μm; A-M Systems) and fixed in place by cyanoacrylate glue. An array of 2 × 4 tetrodes was then inserted and glued on the wall of the stainless steel guide tube. Insulation coats of wire tips were gently removed. Then individual wire tip was soldered to the corresponding pin on a connector. The reference (Nichrome Wire, bare diameter: 50.8 μm, A-M Systems) and ground (copper wire, diameter: 0.1 mm) were also soldered to the corresponding pins. The connector pin was then coated with silica gel. The tetrode was trimmed to an appropriate length immediately before implantation. The impedance of the trimmed tetrode was 0.7–0.8 MΩ at 1 kHz.

For assembly of optetrodes, an extra polymide tubing (inner diameter: 254 μm; wall: 25.4 μm; A-M Systems) was added for optical fiber.

### Surgical implantation of tetrode

All surgeries were conducted under isoflurane anesthesia (3% induction, 1.5–2% maintenance). Body temperature was kept stable throughout the surgery using a heating pad. Aseptic procedures were applied during surgery. Once anesthetized, mice were placed in a stereotaxic apparatus. The scalp was shaved and cleaned with betadine and ethanol, and then a midline incision was made with a scalpel. A spatula was used to clean the skull of all overlying tissue. A craniotomy was made above the AC, and mice were unilaterally implanted in the AC with the tetrodes aimed at the following coordinates: 2.46 mm posterior of bregma, 4 mm lateral to the midline, and 0.65 mm below the cortical surface. A thin layer of tissue gel (3M Vetbond Tissue adhesive, MN, USA) was used to prevent contact of dental acrylic to brain tissue. A stainless steel head plate was then affixed to the skull using screws and dental acrylic, and cement. Finally, dental acrylic and cement were mixed and applied to connect the skull, screws, head plate, and tetrode array. Antibiotic drug (Baytril, 5 mg/kg b.w., Bayer, Whippany, NJ, US) was used for three consecutive days after surgery. After 7–10 days of recovery following surgery, mice were started with a water restriction and behavioral training.

### Recording

The recording began after the mice were well trained. Experiments were conducted in a double-walled, sound-attenuating, and electrically shielded room.

Wideband signals (300–6000 Hz) were recorded using a head-stage amplifier (RHD2132, Intan Technology, CA, USA). Amplified (×20) and digitized (at 20 kHz) neural signals were sent to a USB interface board (RHD2000 Intan Technology) and then sent to the computer for online observation and data storage. Task events such as stimulus presentations and behavioral responses were also

sent to the USB interface board (RHD2000 Intan Technology) and recorded concurrently with the neural data.

## Neural data analysis

Spike sorting was performed using Spike 2 software (version 8, CED, Cambridge, UK). Raw neural signals were band-pass filtered in 300–6000 Hz to remove field potentials. Typically, signals larger than four times the standard deviation of the background noise were considered to be spike events. The detected spike waveforms were then clustered by principal component analysis and a template-matching algorithm. Waveforms with inter-spike intervals of <2 ms were excluded. Only single units with a firing rate higher than 2 Hz were included for further analysis.

Relative spike timing data for a single unit were then obtained for different trials of different conditions and used to construct both raster plots and PSTHs using custom MATLAB scripts. To render PSTHs, all spike trains were first binned at 10 ms and convolved with a smoothing Gaussian kernel ($\delta$ = 100 ms) to minimize the impact of random spike-time jitter at the borders between bins. The baseline period was defined as 0.5 s before the onset of the stimulus. PSTHs in populations were constructed, averaging responses of multiple neurons. As generally observed, behavioral and neuronal results were similar across all relevant animals for a particular testing paradigm. Thus, the data were combined to study population effects.

To quantify the ability to discriminate between the two stimuli, ROC-based analysis was performed. We performed this analysis on consecutive 100 ms epochs. For each epoch, we set 12 threshold levels of activity covering the range of firing rates that followed the two-tone stimuli. For each threshold level, the proportion of trials for the preferred and nonpreferred stimulus showed activity greater than the threshold was calculated. These data were plotted to create a ROC curve. The area under the ROC curve represents the probability that an ideal observer can discriminate between the two stimuli based on the activity. Therefore, a ROC value of 0.5 indicates no difference in the distribution of response in that epoch following the two tones. A value of 1 indicates that the activity following the preferred tone stimulus was always higher than the highest activity following the nonpreferred tone stimulus. A value of 0 indicates that the activity following the preferred tone stimulus was always lower than the lowest activity following the nonpreferred tone stimulus.

To test the significance of each ROC value, we ran a permutation test. This was accomplished by randomly distributing all trials from a neuron into two groups, independent of the actual cues. These groups were nominally called the preferred group and nonpreferred group. The ROC value was calculated from the redistributed data, and the procedure was repeated 5000 times, thereby creating a distribution of ROC values. We then determined wherein the distribution of the actual ROC value lay. Actual value in the top or bottom 5% was defined as significant (i.e., $p < 0.05$).

## Optogenetic experiments

### Virus injection and optical fiber implantation

The virus injection and optical fiber implantation surgery were similar to that for the implant of tetrodes. For optogenetic inactivation experiments, craniotomies (1 mm in diameter) were made bilaterally above the AC. The injecting pipette was pulled from a glass tube (Borosilicate glass capillaries, World Precision Instruments) and back-filled with mineral oil. About 0.5 µl AAV-CaMKIIα-eNpHR3.0-eYFP (Neuron Biotech, Shanghai, China, $7 \times 10^{12}$ particles/ml) or AAV-CaMKIIα-eYFP (Neuron Biotech, $2 \times 10^{13}$ particles/ml) was injected into each hemisphere of AC at the following coordinates: 2.46 mm posterior of bregma, 4 mm lateral to the midline, and 0.65 mm below the cortical surface. Two optic fibers (200 µm in diameter, 0.37 NA) with ceramic ferrule were implanted. Tips of optical fibers were 300 µm over the virus injection sites in AC for each hemisphere, with the coordinates of AP 2.46 mm, ML 4 mm, DV 0.35 mm. Experiments were performed after 0.5–2 months of expression time.

### Optogenetic manipulation during task performance

Experiments were conducted in a double-walled, sound-attenuating room. For optogenetic suppression of the AC during the behavioral task, a green laser (532 nm, 10 mW) was delivered to the AC of both hemispheres through the optic fibers. A splitting optical patch cable was used to connect the two chronically implanted optical fibers (through a ceramic sleeve) to the laser source (R-LG532-200-

A5, RWD, Shenzhen, China). Laser power was controlled by analog inputs sent from the behavior computer through an analog-digital multifunction card (DAQ NI 6363, National Instruments). Laser power at the end of an external fiber was measured with a laser power meter (VLP-2000, RWD).

To verify the effectiveness of the optogenetic method, optetrodes were implanted in AC for simultaneous AC photo-inactivation and recording. The assembly of optetrode was described above. The tip of the tetrode array was extended ~300 μm from the tip of the optical fiber. Light power at the tip of the optical fiber was 10 mW.

## Histology

Once all experiments were completed, mice were deeply anesthetized with sodium pentobarbital (100 mg/kg) and then perfused transcardially with saline followed by 4% paraformaldehyde. The brains were removed from the skull and kept in 4% paraformaldehyde at 4°C overnight, then transferred to PBS. 50 μm coronal slices were cut and placed in PBS. They were then incubated with DAPI for 10–15 min. Slices were washed again in PBS, mounted, and coverslipped. Fluorescence images were then obtained with a confocal microscope.

## Statistical analysis

All statistical analyses were performed in MATLAB. Datasets were tested for normality, and appropriate statistical tests were applied as described in the text (e.g., $t$-test for normally distributed data, Wilcoxon rank-sum test for nonparametric data). Unless otherwise stated, data were reported as mean ± s.e.m.

## Acknowledgements

We thank Song Chang for technical assistance, Dr. Xinjian Li for help in virus injection, Dr. Longnian Lin for help in tetrode manufacturing, Lina Wang for help in imaging experiments, and Dr. Christos Constantinidis for critical comments on the manuscript.

## Additional information

### Funding

| Funder | Grant reference number | Author |
| --- | --- | --- |
| Shanghai Natural Science Foundation | 20ZR1417800 | Jinghong Xu |
| National Natural Science Foundation of China | 31970925 | Liping Yu |
| Shanghai Natural Science Foundation | 19ZR1416500 | Liping Yu |
| National Natural Science Foundation of China | 31400944 | Jinghong Xu |

The funders had no role in study design, data collection and interpretation, or the decision to submit the work for publication.

### Author contributions

Liping Yu, Conceptualization, Software, Formal analysis, Funding acquisition, Investigation, Methodology, Project administration, Writing - review and editing; Jiawei Hu, Chenlin Shi, Formal analysis, Investigation, Methodology; Li Zhou, Maozhi Tian, Jiping Zhang, Methodology; Jinghong Xu, Conceptualization, Software, Supervision, Funding acquisition, Methodology, Writing - original draft, Project administration, Writing - review and editing

### Author ORCIDs

Jinghong Xu (iD) https://orcid.org/0000-0002-2864-4196

## Ethics

Animal experimentation: All experiments were performed in strict accordance with the recommendations in the Guide for the Care and Use of Laboratory Animals of the US National Institutes of Health. The protocol was approved by the Animal Care and Use Committee of East China Normal University, Shanghai, China (m20160302).

## Decision letter and Author response

Decision letter https://doi.org/10.7554/eLife.64457.sa1
Author response https://doi.org/10.7554/eLife.64457.sa2

# Additional files

## Supplementary files

• Transparent reporting form

## Data availability

Data deposited in Dryad Digital Repository, accessible here: https://doi.org/10.5061/dryad.8gtht76nf.

The following dataset was generated:

| Author(s) | Year | Dataset title | Dataset URL | Database and Identifier |
|---|---|---|---|---|
| Yu L | 2021 | The causal role of auditory cortex in auditory working memory | https://doi.org/10.5061/dryad.8gtht76nf | Dryad Digital Repository, 10.5061/dryad.8gtht76nf |

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
