## [Decision Letter]

**Acceptance summary:**

This study elegantly shows based on an ambitious set of optogenetic experiments that auditory cortex is necessary for encoding and transient maintenance of sound information in an auditory working memory task. Moreover, electrophysiological recordings during the task show that auditory cortex activity is modulated by the working memory context, further corroborating the idea that its activity is important, at least transiently, for working memory. These results complement current theories that involve mainly prefrontal cortex in working memory, by suggesting that a larger part of the cortical network, including the relevant sensory areas, are actually involved in short-term sensory information storage.

**Decision letter after peer review:**

Thank you for submitting your article "The causal role of auditory cortex in auditory working memory" for consideration by *eLife*. Your article has been reviewed by 3 peer reviewers, including Brice Bathellier as the Reviewing Editor and Reviewer #1, and the evaluation has been overseen by Andrew King as the Senior Editor. The following individual involved in review of your submission has agreed to reveal their identity: Yves Boubenec (Reviewer #3).

The reviewers have discussed the reviews with one another and the Reviewing Editor has drafted this decision to help you prepare a revised submission.

Summary:

The authors examined whether and how auditory cortex (AC) is important for memory maintenance in an auditory Working Memory (WM) task, using a delayed non-match to sample design and go/no-go response as behavioral readout. Electrophysiological recordings in AC showed WM-encoding activity early in the delay period, but not in the late delay period. Optogenetic suppression of neural activity in AC during the early, but not late, delay period impaired WM performance. Optogenetic suppression of AC activity in a delayed response task did not modulate performance. The causal evidence for the involvement of AC in the early delay period is clear, while the irrelevance of AC in the later delay period is also clearly demonstrating the limitation of AC involvement in WM. The results are interesting to the WM field and beyond because they suggest a transient storage of WM in a primary sensory cortical area. However, a few more experiments, more careful interpretation and further discussion are required to reinforce and refine this important claim.

Essential revisions:

1. The task used will be very useful for the community and therefore the authors should provide much more detail about the protocol and the indicators of task performance. They should show population learning curves, fraction of mice learning the task, mean duration to reach 80% correct, and, for all tasks, the mean lick PSTH for all delay durations. This is important because mice tend to produce impulsive licking to sounds, and one must be able to evaluate in the context of this task whether this aspect is under experimental control. The authors should also mention in the protocol if (and how) impulsive licking before the test sound is punished.

2. A major concern is the interpretation of early involvement of AC in the task. If AC is only important for the early delay period, then one can argue that AC is only reflecting the residual auditory trace, or immediate memory, which is profoundly different from active memory maintenance to guide behavior performance. The authors should tone down their claim about the importance of AC in the working memory process and more carefully discuss it. In particular, the authors should address the counter-argument about the irrelevance of sensory regions in WM, for example Xu (2018) Trends in Cognitive Sciences 2018 (doi: 10.1016/j.tics.2017.12.008).

3. Active WM maintenance requires resistance to distractors presented during delay period. In line with the previous point, the authors should examine whether AC is important if a distractor is applied during the early delay period to mask residual sensory activity. Optogenetic suppression data obtained in this distractor condition should clearly show whether AC is important for maintenance of WM or only reflecting residual sensory information.

4. Surprisingly, the authors have not tested whether sound responses in AC are necessary to perform the task as no inactivation was performed before sound onset. It is important to determine how important AC is during sound presentation to evaluate whether what is assumed to reflect working memory corresponds to transmission of auditory information occurring just before in AC or something else.

5. The lack of delayed response during passive listening shows that this sustained activity cannot be due to a pure acoustic response. In addition, the inactivation experiments suggest a causal role for this activity in working memory. However, it is not clear whether this delayed activity is locked to the stimulus offset, or instead whether its temporal dynamics are independent from the stimulus offset, and instead relative to the stimulus onset. In other words, working-memory processes could be triggered by the end of the sound, or could instead be a process whose dynamics unfold over several hundreds of milliseconds and which would emerge only later on (coincidentally after stimulus offset). The short duration of the first sound token does not allow one to disentangle these two possibilities. A way to test for these hypothesis would be to repeat the experiment with variable sound duration, in order to see whether the sustained activity would shift with later stimulus offset.

6. The authors should more carefully review literature about sensory cortices in WM. For example, Colombo et al., Science, 1990 (doi: 10.1126/science.2296723) and Zhang et al., *eLife* , 2019 (DOI: 10.7554/*eLife*.43191) should be cited, which showed causal evidence of sensory cortices in WM tasks.

---

## [Author Response]

Essential revisions:1. The task used will be very useful for the community and therefore the authors should provide much more detail about the protocol and the indicators of task performance. They should show population learning curves, fraction of mice learning the task, mean duration to reach 80% correct, and, for all tasks, the mean lick PSTH for all delay durations. This is important because mice tend to produce impulsive licking to sounds, and one must be able to evaluate in the context of this task whether this aspect is under experimental control. The authors should also mention in the protocol if (and how) impulsive licking before the test sound is punished.

Thank you for the suggestions. We reviewed the behavioral data during training and found that it took an average of 29 (25-35) days of training to reach the level of behavioral performance >80% correct. The performances of all mice while learning the WM task were shown in Figure 1—figure supplement 1a. In total, we trained 58 mice to perform the task and removed 17 mice from the study before data collection because they failed to show apparent progress like others in stage 3 of training, during which the two match and two non-match trials were presented randomly. Finally, the majority (71%, 41/58) of mice could learn the task.

The mean lick PSTHs of well-trained mice while performing the WM task, delayed go/no-go task, and go/no-go task were shown in Figure 1—figure supplement 1b, c, d, e, and f. Once the mice were well-trained, they readily performed the task with little licking during the delay period, and these lickings were not punished.

We have added this information in the section of *Materials and methods* (see page 20, the second paragraph; and page 22, the second paragraph).

2. A major concern is the interpretation of early involvement of AC in the task. If AC is only important for the early delay period, then one can argue that AC is only reflecting the residual auditory trace, or immediate memory, which is profoundly different from active memory maintenance to guide behavior performance. The authors should tone down their claim about the importance of AC in the working memory process and more carefully discuss it. In particular, the authors should address the counter-argument about the irrelevance of sensory regions in WM, for example Xu (2018) Trends in Cognitive Sciences 2018 (doi: 10.1016/j.tics.2017.12.008).

In the present study, we observed a temporal specific effect of AC suppression in the WM task. Only optogenetic suppression of AC activity during the stimulus epoch and early delay period caused a substantial reduction in WM performance. This result was consistent with the electrophysiology result showing that AC neurons exhibited elevated activity during stimulus presentation and early delay period. These results indicated that although AC may not be involved in WM storage during the whole delay period, it is nevertheless crucial for WM tasks in the initial encoding of auditory information, maintaining the memory trace for a limited time, and then transferring this information for further WM storage elsewhere, probably in posterior parietal cortex (PPC) and prefrontal cortex (PFC).

There is an ongoing debate concerning the necessity of the delay-period activity of sensory cortices in WM. The essential theories which argue for the importance of sensory cortex are based on recording and imaging experiments showing that during the memory period, neuronal activity in sensory cortices can be maintained and reflect the identity of the remembered stimulus (Fuster and Jervey, 1981; Gayet et al., 2018; Harrison and Tong, 2009; Mendoza-Halliday et al., 2014; Pasternak and Greenlee, 2005; Scimeca et al., 2018). Furthermore, perturbation of neural activity in sensory cortices can impair WM performance (Colombo et al., 1990; Harris et al., 2002; Zhang et al., 2019). The unessential theories are based on observations that when distractors were applied during the delay period, the sustained neural activity and WM decoding were no longer present in sensory cortices. Meanwhile, the presence of the distractor had no impact on behavioral performance (Bettencourt and Xu, 2016; Miller et al., 1996). This suggests that sensory cortices may not be essential for memory maintenance, and the WM-related activities observed in sensory cortices may largely reflect feedback signals indicative of the storage of WM content elsewhere in the brain, probably in the posterior parietal cortex (PPC) and prefrontal cortex (PFC) (Leavitt et al., 2017; Xu, 2017; Xu, 2018).

The reviewer raised the critical question about whether the early delay-period activity of AC reflected the active memory maintenance or only the residual auditory trace. To further study the role of AC early delay-period activity in WM, we introduced a noise distractor (20-20,000Hz, 200ms, 60dB) during the early delay period (300-500ms) and test whether AC was involved in memory maintenance even following the distractor. If the early delay-period activity of AC is essential for WM, the neural activity should be able to maintain information following the distractor. Our result showed that suppressing the early delay-period activity of AC after the distractor resulted in performance impairment (Figure 6b). Therefore, AC is essential for active WM maintenance during the early delay period. The early delay-period activity observed in AC reflected the active memory maintenance not the residual auditory trace. Generally, our present data support the idea that sensory cortices participate in sensory information maintenance in WM task. We have added this information in the *Results section (see page 10, last paragraph).*

*Once again, we appreciate the suggestions and have explained this in the Discussion* part of the manuscript (see page 13, the first and second paragraph).3. Active WM maintenance requires resistance to distractors presented during delay period. In line with the previous point, the authors should examine whether AC is important if a distractor is applied during the early delay period to mask residual sensory activity. Optogenetic suppression data obtained in this distractor condition should clearly show whether AC is important for maintenance of WM or only reflecting residual sensory information.

A hallmark of active WM maintenance is the resistance against distractors during the delay period (Baddeley, 2012; Bettencourt and Xu, 2016; Miller et al., 1996). As stated above (please see the response to Q2), we did a new experiment where a noise distractor was added into the delay period (300-500ms). We found that mice could quickly adapt to the WM task with noise distractor despite the initial drop in performance (Figure 6a). The behavioral performance in the presence of noise distractor appears no worse than that in the simple WM task. We then optogenetically suppressed the AC activity in a specific delay period (500-800ms, just following the noise distractor, see Figure 6b). The result showed that optogenetic suppression resulted in impairment in task performance (Figure 6b). This new evidence further indicates that AC activity is critical for active early maintenance of the auditory information in WM task.

Now we added this result as an additional figure to the manuscript (Figure 6) and also added this information in the *Results* part of the manuscript (See Page 10, last paragraph). Thank you for the suggestion.

4. Surprisingly, the authors have not tested whether sound responses in AC are necessary to perform the task as no inactivation was performed before sound onset. It is important to determine how important AC is during sound presentation to evaluate whether what is assumed to reflect working memory corresponds to transmission of auditory information occurring just before in AC or something else.

The reviewer raises the question about the role of AC auditory response in WM. To answer this question, we optogenetically suppressed the AC activity during the stimulus epoch. We found that optogenetic suppression resulted in dramatic behavioral performance decline. This result suggests that the auditory information encoded by AC during the stimulus epoch is crucial for WM task. The WM information maintained in AC during the early delay period may reflect the prolonging of the representations formed during sensory processing.

We added this as an additional supplementary figure to the manuscript (Figure 3—figure supplement 1) and added this information in the *Results* part of the manuscript (See Page 8, the second paragraph). Thank you for the suggestion.

5. The lack of delayed response during passive listening shows that this sustained activity cannot be due to a pure acoustic response. In addition, the inactivation experiments suggest a causal role for this activity in working memory. However, it is not clear whether this delayed activity is locked to the stimulus offset, or instead whether its temporal dynamics are independent from the stimulus offset, and instead relative to the stimulus onset. In other words, working-memory processes could be triggered by the end of the sound, or could instead be a process whose dynamics unfold over several hundreds of milliseconds and which would emerge only later on (coincidentally after stimulus offset). The short duration of the first sound token does not allow one to disentangle these two possibilities. A way to test for these hypothesis would be to repeat the experiment with variable sound duration, in order to see whether the sustained activity would shift with later stimulus offset.

In the present study, we observed that after the offset of the 200ms sample stimulus, the AC neurons continued to exhibit activity for 800ms into the delay period, 1000ms from the stimulus onset in total. However, it is not clear whether the temporal dynamics of delay-period activity is relative to the stimulus onset or offset. The reviewer makes a good suggestion of increasing the stimulus duration to test the two possibilities. Thus, we increased the stimulus duration from 200ms to 300ms and 400ms and then tested whether the sustained activity would shift with later stimulus offset. The AC activity was recorded from a subgroup of mice (n=4) while performing the auditory WM task with the stimulus duration of 300ms or 400ms. The averaged population firing rates showed that the neurons exhibited phasic responses during the auditory sample stimulus presentation. After the offset of the sample stimulus, the neurons continued to exhibit elevated activity for 700ms into the delay period in the 300ms stimulus duration task, and exhibit elevated activity for 600ms into the delay period in the 400ms stimulus duration task. Together with the result from the 200ms sample stimulus duration task, which showed that the neurons continued to exhibit elevated activity for 800ms into the delay period, all of these results showed that AC neurons exhibited elevated activity for 1000ms from the stimulus onset, regardless of the duration of the sample stimulus. These results suggested that the temporal dynamics of the delay-period activity in AC might be relative to the stimulus onset rather than the offset.

We added this as an additional figure to the manuscript (Figure 2—figure supplement 1) and added this information in the *Results* (See Page 7, the second paragraph). Thank you for the suggestion.

6. The authors should more carefully review literature about sensory cortices in WM. For example, Colombo et al., Science, 1990 (doi: 10.1126/science.2296723) and Zhang et al., eLife , 2019 (DOI: 10.7554/eLife.43191) should be cited, which showed causal evidence of sensory cortices in WM tasks.

We have carefully reviewed literature and cited more necessary literature about the sensory cortices in WM in the proper place of the manuscript.

References:

Baddeley A. 2012. Working memory: theories, models, and controversies. Annu Rev Psychol 63:1-29.Bettencourt KC, Xu Y. 2016. Decoding the content of visual short-term memory under distraction in occipital and parietal areas. Nat Neurosci 19:150-157.Colombo M, D'Amato MR, Rodman HR, Gross CG. 1990. Auditory association cortex lesions impair auditory short-term memory in monkeys. Science 247:336-338.Fuster JM, Jervey JP. 1981. Inferotemporal neurons distinguish and retain behaviorally relevant features of visual stimuli. Science 212:952-955.Gayet S, Paffen CLE, Van der Stigchel S. 2018. Visual Working Memory Storage Recruits Sensory Processing Areas. Trends Cogn Sci 22:189-190.Harris JA, Miniussi C, Harris IM, Diamond ME. 2002. Transient storage of a tactile memory trace in primary somatosensory cortex. J Neurosci 22:8720-8725.Harrison SA, Tong F. 2009. Decoding reveals the contents of visual working memory in early visual areas. Nature 458:632-635.Leavitt ML, Mendoza-Halliday D, Martinez-Trujillo JC. 2017. Sustained Activity Encoding Working Memories: Not Fully Distributed. Trends Neurosci 40:328-346.Mendoza-Halliday D, Torres S, Martinez-Trujillo JC. 2014. Sharp emergence of feature-selective sustained activity along the dorsal visual pathway. Nat Neurosci 17:1255-1262.Miller EK, Erickson CA, Desimone R. 1996. Neural mechanisms of visual working memory in prefrontal cortex of the macaque. J Neurosci 16:5154-5167.Pasternak T, Greenlee MW. 2005. Working memory in primate sensory systems. Nat Rev Neurosci 6:97-107.Scimeca JM, Kiyonaga A, D'Esposito M. 2018. Reaffirming the Sensory Recruitment Account of Working Memory. Trends Cogn Sci 22:190-192.Xu Y. 2017. Reevaluating the Sensory Account of Visual Working Memory Storage. Trends Cogn Sci 21:794-815.Xu Y. 2018. Sensory Cortex Is Nonessential in Working Memory Storage. Trends Cogn Sci 22:192-193.Zhang X, Yan W, Wang W, Fan H, Hou R, Chen Y, Chen Z, Ge C, Duan S, Compte A, Li CT. 2019. Active information maintenance in working memory by a sensory cortex. *eLife* 8:e43191.